# The Concurrent Validity and Test-Retest Reliability of Possible Remote Assessments for Measuring Countermovement Jump: My Jump 2, HomeCourt & Takei Vertical Jump Meter

**Gary Chi-Ching Chow * , Yu-Hin Kong and Wai-Yan Pun**

Department of Health and Physical Education, The Education University of Hong Kong, 10 Lo Ping Road, Tai Po, Hong Kong
* Correspondence: ccchow@eduhk.hk; Tel.: +852-2948-8849; Fax: +852-2948-7848

**Featured Application: The same tool could be used repeatedly to track the changes in CMJ performance. Average jump heights should be analyzed. Practitioners and sports professionals without extensive knowledge of assessment could self-administer CMJ tests using these devices.**

**Abstract:** Mobile applications and portable assessments make remote self-assessment of the countermovement jump (CMJ) test possible. This study aimed to investigate the concurrent validity and test–retest reliability of three portable measurement systems for CMJ. Thirty physically active college students visited the laboratory twice, with two days in between, and performed three jumps each day. All jumps were recorded by My Jump 2, HomeCourt, and the Takei Vertical Jump Meter (TVJM) simultaneously. Results indicated significant differences among the three systems ($p < 0.01$). Home-Court tended to present the highest jump height mean value ($46.10 \pm 7.57$ cm) compared with TVJM ($42.02 \pm 8.11$ cm) and My Jump 2 ($40.85 \pm 7.86$ cm). High concurrent validities among assessments were found ($r = 0.85$–$0.93$). Good to excellent reliability of jump assessments was demonstrated ($ICC_{3,1} = 0.80$–$0.96$). Reliable coefficients of variation were shown in all measurements (2.58–5.92%). Significant differences were revealed among the three apparatuses while they demonstrated high intra-device test–retest reliability. TVJM was the most reliable, and average jump heights were recommended for analysis.

**Keywords:** countermovement jump; reliability; remote assessment; technology; validity



## 1. Introduction

Coronavirus Disease 2019 has limited onsite sports training and assessments. The quarantine is more detrimental than the usual transition phase and has induced significant unfavorable effects on physical conditioning [1]. The countermovement jump (CMJ) test is a popular and valid test to monitor neuromuscular status in sports performance. Indeed, jump height is sensitive to acute fatigue [Hedges' g = −0.27 (95% CI: −0.48, −0.05)] and chronic adaptation [Hedges' g = 0.37 (95% CI: 0.32, 0.43)] [2]. Jump height has also been applied to determine physical conditioning in different athletic populations during the pandemic [1,3,4]. Comparing other assessments, for instance, sprint tests, CMJ only requires limited space. Its simplicity and effectiveness have been widely recognized by 61% of sports practitioners in individual and team sports as well as fundamental strength training programs [5]. The impaired CMJ performance after a match helped determine soccer players' readiness and implicate recovery [6]. Moreover, jump height is an indicator that is easy to understand and communicate between practitioners and researchers with a comparable coefficient of variation (CV) and sensitivity, similar to other CMJ parameters, for example, peak power, so the gap between researching and practicing would be easier to bridge.

Rago et al. [7] reviewed the concurrent validity, test–retest reliability, and applications of portal CMJ devices for field testing. The authors found that applying some measurements to athletes in a remote setting was not feasible due to the equipment's cost and size. In the last decade, mobile applications and cost-friendly portable devices were developed for the sports industry. More importantly, it has been necessary to scientifically validate the tools for collecting physiological and performance data. McMahon et al. [8] commented that without validation with the criterion method of assessing CMJ jump height, the new measurement devices should not be assumed to yield valid values. For instance, the Takei Vertical Jump Meter (TVJM), which is a portable wire-type linear encoder and feasible to use in remote assessment, was criticized for its accuracy due to the lack of validation study.

Mobile applications and portable apparatus for assessments have the potential to foster remote performance testing for athletes. Indeed, the practice of sports coaching has been changed by the COVID-19 pandemic, and it necessitates the rapid development of remote monitoring of functional performance in sports. Numerous cheaper, non-invasive, and safer instruments have recently been invented to serve this purpose. However, to apply emerging mobile applications in the practical sports field, accuracy and consistency need to be examined. Instead of a general comparison with an expensive and hardly transportable force platform, a concurrent validity study is acceptable to determine the validity of new measurements compared with a well-established test and a previously validated system [9,10]. My Jump 2 achieved strong concurrent validity in comparison with the force platform (ICC > 0.99) [11,12] and was recognized as a highly useful instrument for field-based CMJ assessments. It is valid and feasible to adopt My Jump 2 as a criterion instrument for a validity–reliability study in developing vertical jump measuring devices. Considering the availability of scientific equipment, Till et al. [13] responded that researchers and practitioners might adopt validated measurements where possible, with clear data presentation on the population-specific typical error and minimal detectable change (MDC) reliability data. Previous studies revealed the reliability of jump assessments but did not provide a meaningful value for practitioners [11,14,15]. MDC, a value calculated by the standard error of measurement (SEM), is suggested to provide in this test–retest study [16].

Therefore, this study evaluated the concurrent validity and test–retest reliability of three portable measurement systems for CMJs, i.e., My Jump 2, HomeCourt, and the TVJM. The findings could be developed as a remote physical conditioning assessment in home-based training.

## 2. Materials & Methods

A cross-sectional study design was adopted to compare the jump height of CMJs measured by My Jump 2, HomeCourt, and the TVJM. Participants were tested in two sessions separated by two days so that within-day and between-day reliability could be examined. This study design is similar to other concurrent validity and reliability studies [7,9,11,12,15,17,18].

### 2.1. Participants

Thirty physically active college students (14 males and 16 females; age = 23.03 ± 1.67 years; height = 167.86 ± 6.27 cm; weight = 61.89 ± 9.82 kg; physical activity level = 4614.52 ± 1332.69 MET-minutes/week) were recruited via emails with an attached information sheet and a consent form. The inclusion criteria were: (1) age 18 or above, (2) classified as a highly active individual according to the time they spent on physical activity in the last seven days (MET-minutes/week) collected by the International Physical Activity Questionnaire Short Form (IPAQ-SF) [19], and (3) able to perform CMJs correctly. There was no exclusion criterion. The study was conducted in accordance with the Declaration of Helsinki, and the protocol was approved by The Human Research Ethics Committee of the Education University of Hong Kong (Ref. no.: 2020-2021-0362). Participants were fully informed that they preserved the right to decline to join or withdraw

from participation at any time without any consequence. They provided informed consent before the test started. No injury was reported throughout the data collection.

### *2.2. Procedures*

Data were collected in the Human Performance Laboratory of the Education University of Hong Kong. Participants were advised to wear tight-fitting sportswear and completed two 40-min sessions with two days of rest in between. A similar time for both days (±1 h) was arranged. In the first session, written consent was obtained after a briefing. Demographic and anthropometric measurements were followed. Afterward, participants completed the IPAQ-SF with the help of the principal researcher. A 10-min standardized warm-up that included specific warm-up CMJs and familiarization prior to experimental testing was provided. Three CMJ trials were then performed. The second session measured three more CMJs after the standardized warm-up. All jumps were recorded by My Jump 2 (video frame identification mobile application), HomeCourt (artificial intelligence [AI] motion tracking mobile application), and the TVJM simultaneously. Two-minute rest were arranged between trials to minimize potential fatigue [18]. All participants in this study completed the two testing sessions in the same week (Figure 1).

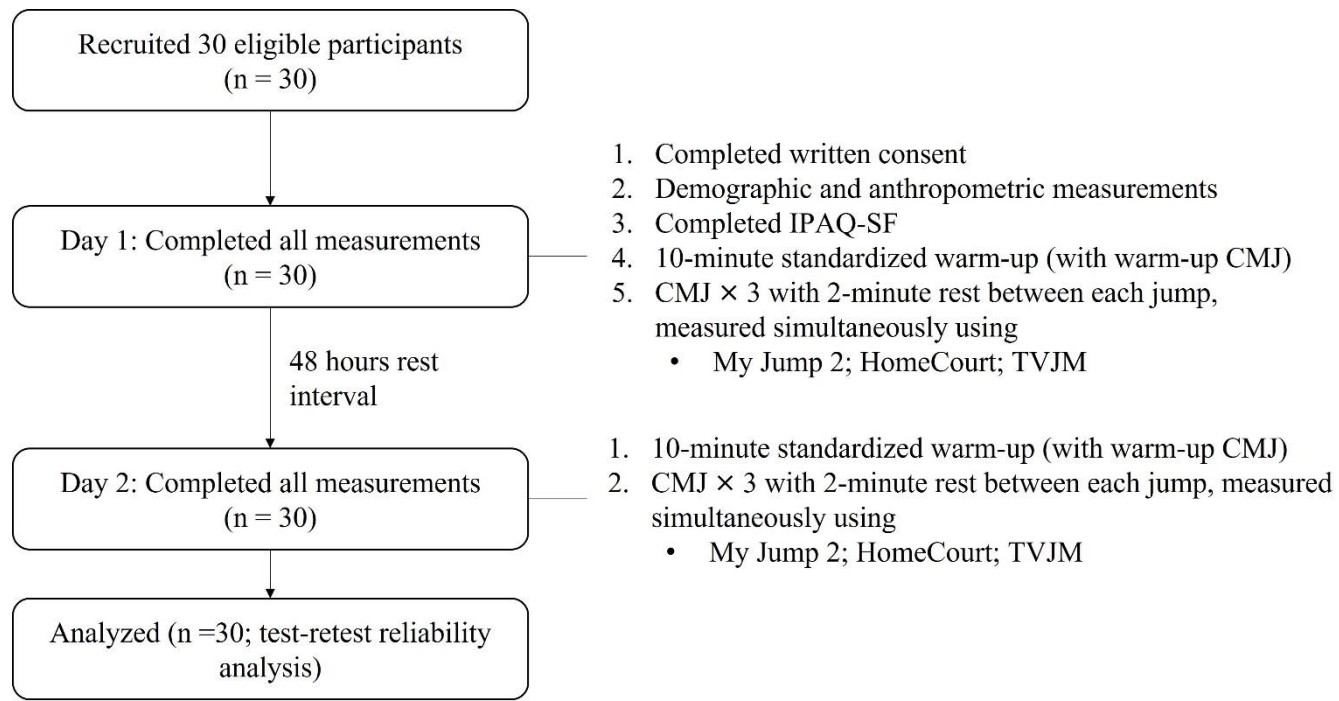

**Figure 1.** Study flow chart. Abbreviations: CMJ: Countermovement jump; TVJM: Takei Vertical Jump Meter.

### *2.3. CMJ*

Vertical CMJs were measured simultaneously using the three assessment devices. The participants began by standing with both feet flat on hard ground with their hands on hips to eliminate the influence of arm movement. They squatted down until approximately 90° knee flexion as quickly as possible and then jumped upward explosively [20]. The participants then landed in a similar fully extended posture. A review [21] found that three attempts were the highest preference (76%) among the 62 reviewed validity–reliability studies in CMJ. Participants with highly active backgrounds were able to perform the CMJ correctly. In each testing session, warm-up CMJs were given before three maximal CMJs. The test–retest reliability of the CMJ test is very high (ICC: 0.98) [22]. Verbal encouragement was provided, while feedback related to the jumping skills was omitted. A total of six

maximal CMJs in two days were evaluated. The average performance of all three CMJs each day was used for the reliability analysis.

### 2.4. Measurements

The two iPads (6th generation, iOS 14.4), one with My Jump 2 and another one with HomeCourt and the TVJM, were used to assess jump height. The jumping height (cm) of each jump was recorded and used for data analysis. Participants stood on the TVJM's rubber mat with the belt around their waist and hands on their hips. Two iPad devices were placed on two foldable tablet stands without protective cases and were placed three meters in front of the participant side by side (Figure 2). The foldable tablet stands were set with a standardized angle of elevation at 75° in order to capture the entire body during the jumps using both mobile applications. This equipment setup was determined by the researchers to minimize measurement error.

My Jump 2 is a video frame identification mobile application that provides a portable approach for measuring vertical jump performance. It uses the high-speed camera (at least ≥120 fps) of smart devices and adopts the HSC–Kinovea method to record the flight time and calculate jump height [17]. Almost perfect agreement (ICC > 0.99) and correlation ($r$ > 0.99) in CMJ height were found with the force platform [11]. The inter-rater reliability is also close to perfect (ICC = 0.97) [14]. Previous studies suggested that My Jump 2 was a valid and reliable tool for evaluating a wide range of populations (elderly: ICC = 0.95 [23]; children: ICC > 0.89 [9]; trained athletes: ICC > 0.97 [18]), showing its ability to assess participants with various CMJ performances. In this study, the My Jump 2 application was installed on an iPad. Before the jump assessment, (i) leg length from the greater trochanter to the lateral malleolus, (ii) from the lateral malleolus to the tip of toes, and (iii) vertical distance between the greater trochanter and the ground in a parallel squat position were measured and entered in the application for calculation. The jumps were recorded by the 120 Hz high-speed camera of the iPad with a quality of 720 p. The frames of take-off and landing were marked manually by one researcher, and the jump height was calculated automatically using this equation:

$$h = t^2 \times 1.22625$$

where $h$ was the jump height in meters, and $t$ was the flight time in seconds [10]. The record would be shown on the screen, and the smallest measurement unit was 0.01 cm.

HomeCourt (NEX Team Inc., San Jose, CA, USA, and Hong Kong) was a mobile application that was originally developed for tracking basketball shooting. Additionally, the latest iOS version includes the features of augmented reality and body segments for tracking a person's movements through the camera of the smart devices. This new addition empowers the development of HomeCourt to recognize objects and track human and ball movements. Apart from tracking basketball shooting, it allows users to assess vertical jump height using its AI. Each jump record would be shown on the screen, and the smallest measurement unit was 1 cm. Before the jump assessment, the application automatically measured users' wingspans, standing reach, and seated height for reference, which took about 30 s.

The TVJM (T.K.K. 5406 Jump MD, Takei Scientific Instruments Co., Ltd., Niigata, Japan) is a linear position transducer with a wire-type linear encoder. Till and his colleagues [24–26] first used this instrument for CMJ assessment and reported high reliability ($r$ = 0.90). In this study, participants stood on the Takei rubber mat (approximately 380 mm in diameter and 3 mm in thickness, weighted at 0.4 kg) with a belt around the waist. The belt weighs only 0.2 kg and is connected to the mat with a measurement cord which initially represents the distance between the waist and the ground. The cord elongated during a jump. The height of each jump was displayed on the LCD screen on the belt. The smallest measurement unit was 1 cm, and its error was up to ±2 cm.

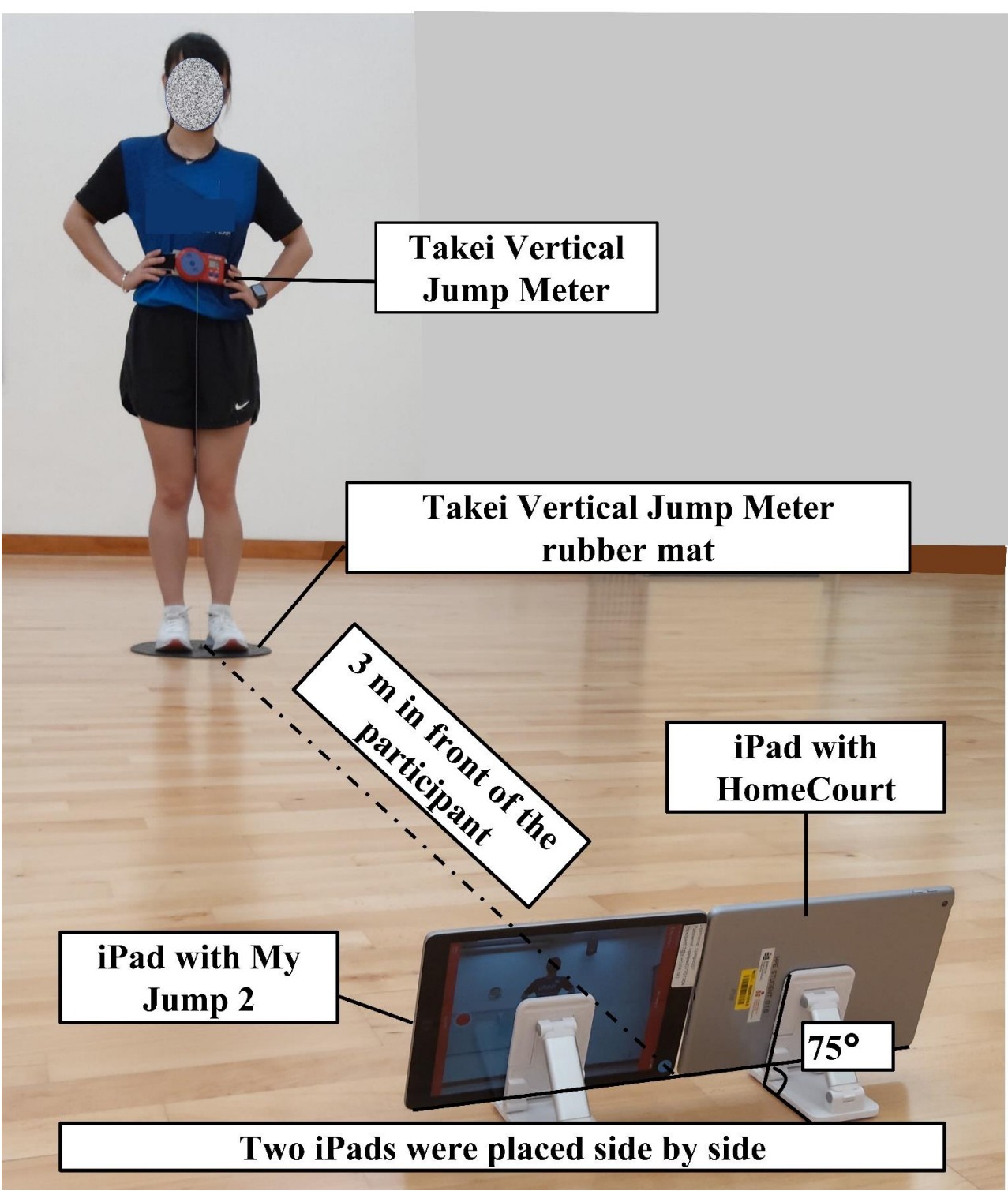

**Figure 2.** Measurements placement for the CMJ reliability test.

### 2.5. Statistical Analyses

All jumps were combined to provide a total of 180 data points. Descriptive statistics were computed for all variables and reported as mean ± SD. The Shapiro–Wilk test and histograms were used to check the normality of the data. An ANOVA with repeated measures was used to indicate differences among the three measurement systems. Effect size (ES) was interpreted using Cohen's *d* according to Hopkins [27], i.e., trivial (<0.2), small (0.2–0.6), moderate (0.6–1.2), and large (1.2–2.0). Pearson correlation coefficients (*r*) were



also calculated to reveal the concurrent validity among the three systems, while My Jump 2 was the reference standard in this study. A trivial effect size (ES < 0.2) and a large correlation ($r \geq 0.90$) indicated sufficient concurrent validity. Multiple intraclass correlation coefficient (ICC) analysis, single measure- two-way mixed, and absolute agreement parameters ($ICC_{3,1}$) were used to assess within-day and between-day reliability [28], which could be classified as moderate (0.50–0.75), good (>0.75) and excellent (>0.9) [16]. CVs were used to determine the magnitude of measurement error, which were classified as good (<5%), moderate (5–10%), and poor (>10%). $ICC \geq 0.75$ and $CV \leq 10\%$ were defined as sufficient reliability. Moreover, the SEM and MDC were also calculated. IBM Statistical package SPSS version 26.0 (Armonk, NY, USA) and Microsoft Excel 365 (Microsoft Corp., Redmond, WA, USA) were used. The level of statistical significance was set at 0.05 (two-tailed).

## 3. Results

Referring to all trials by the thirty participants in two sessions, an ANOVA with repeated measures indicated significant differences among the three devices, $F(2, 358) = 181.98$, $p < 0.01$. HomeCourt tended to present the highest jump height mean value ($46.10 \pm 7.57$ cm) compared with the TVJM ($42.02 \pm 8.11$ cm) and My Jump 2 ($40.85 \pm 7.86$ cm). Post hoc analysis with a Bonferroni adjustment revealed that the jump heights reported in My Jump 2 were statistically lower than that in the TVJM by 1.17 cm (95% CI: 0.63, 1.70, $p < 0.01$) and that in HomeCourt by 5.25 cm (95% CI: 4.49, 6.01, $p < 0.01$). Descriptive data relative to the reference standard measure, My Jump 2, was presented in Table 1. High Pearson's correlations among the three measurement systems ($r = 0.85–0.93$, $p < 0.01$) revealed good to excellent concurrent validity (Table 2).

**Table 1.** Summary statistics of jump heights across three devices in active college students.

| | CMJ Height (cm) Mean ± SD | | |
|---|---|---|---|
| | My Jump 2 (I) | HomeCourt (J) | TVJM (K) |
| Overall | 40.85 ± 7.86 | 46.10 ± 7.57 | 42.02 ± 8.11 |
| Day 1 | 40.91 ± 7.75 | 45.58 ± 7.50 | 41.80 ± 8.02 |
| Day 2 | 40.79 ± 7.44 | 46.62 ± 6.83 | 42.23 ± 8.05 |
| Overall Mean Difference (J–I) (95% CI) ES (*d*) | | 5.25 (4.49, 6.01) * *d* = 0.68 | |
| Overall Mean Difference (K–I) (95% CI) ES (*d*) | | | 1.17 (0.63, 1.70) * *d* = 0.15 |

Notes: * $p < 0.01$ (2-tailed); CMJ: Countermovement jump; CI: Confidence interval; ES: Effect size; TVJM: Takei Vertical Jump Meter.

**Table 2.** Pearson's correlations among devices of My Jump 2, HomeCourt, and TVJM.

| | My Jump 2 | HomeCourt | TVJM |
|---|---|---|---|
| My Jump 2 | 1 | 0.85 * | 0.93 * |
| HomeCourt | | 1 | 0.85 * |
| TVJM | | | 1 |

Notes: * $p < 0.01$ (2-tailed); TVJM–Takei Vertical Jump Meter.

Good to excellent reliability of both within-day ($ICC_{3,1} = 0.82–0.95$) and between-day ($ICC_{3,1} = 0.89–0.97$) assessments was demonstrated (Tables 3 and 4). Reliable CVs were shown in all measurements (within-day: 3.82–6.45%; between-day: 2.58–4.31%). The TVJM demonstrated the highest reliability [within-day day 1: $ICC_{3,1} = 0.92$ (95% CI: 0.85, 0.96); within-day day 2: $ICC_{3,1} = 0.95$ (95% CI: 0.91, 0.97); between-day $ICC_{3,1} = 0.97$ (95% CI: 0.93, 0.98)] compared with the other two devices. The SEM and MDC of the TVJM in between-day comparison were 1.46 cm and 4.05 cm, respectively.

**Table 3.** Within-day test–retest reliability of CMJ measurement devices.

| | Day 1 | | | | Day 2 | | | |
|---|---|---|---|---|---|---|---|---|
| | ICC$_{3,1}$ (95% CI) | CV% | SEM | MDC (cm) | ICC$_{3,1}$ (95% CI) | CV% | SEM | MDC (cm) |
| My Jump 2 | 0.86 (0.72–0.93) | 6.45 | 3.07 | 8.51 | 0.88 (0.79–0.94) | 5.92 | 2.72 | 7.55 |
| HomeCourt | 0.83 (0.71–0.91) | 6.14 | 3.34 | 9.26 | 0.82 (0.68–0.90) | 5.52 | 3.14 | 8.71 |
| TVJM | 0.92 (0.85–0.96) | 4.70 | 2.32 | 6.43 | 0.95 (0.91–0.97) | 3.82 | 1.82 | 5.05 |

Notes: TVJM: Takei Vertical Jump Meter; ICC: Intraclass correlation coefficient; CV: coefficient of variation; SEM: standard error of measurement; MDC: minimal detectable change.

**Table 4.** Between-day test–retest reliability of CMJ measurement devices.

| | CMJ [Mean ± SD (cm)] | | ICC$_{3,1}$ (95% CI) | CV% | SEM | MDC (cm) |
|---|---|---|---|---|---|---|
| | Day 1 | Day 2 | | | | |
| My Jump 2 | 40.91 ± 7.75 | 40.79 ± 7.44 | 0.93 (0.87–0.97) | 3.68 | 1.95 | 5.41 |
| HomeCourt | 45.58 ± 7.50 | 46.62 ± 6.83 | 0.89 (0.77–0.94) | 4.31 | 2.42 | 6.71 |
| TVJM | 41.80 ± 8.02 | 42.23 ± 8.05 | 0.97 (0.93–0.98) | 2.58 | 1.46 | 4.05 |

Note: CMJ: Countermovement jump; TVJM: Takei Vertical Jump Meter; CI: confidence interval; CV: coefficient of variation; ICC: intraclass correlation coefficient; MDC: minimal detectable change; SD: standard deviation; SEM: standard error of measurement; TVJM: Takei Vertical Jump Meter.

## 4. Discussion

This study was a timely and novel validation study to examine the concurrent validity and test–retest reliability of the three selected possible remote assessments for CMJ, namely My Jump 2, a popular and valid mobile application, and HomeCourt, a mobile application with AI, and the TVJM, a portable jump assessment tool. So as to provide references to practitioners on the meaningful changes in performance, this study provided the MDC of each device, which is the minimal number of measured changes that fell outside measurement error. This study provided evidence and information to researchers and practitioners to select devices for CMJ remote assessment. All selected devices showed good to excellent test–retest reliability (ICC$_{3,1}$: 0.82–0.97). In general, variabilities in CMJ outputs were low (CV: 2.58–6.45%), and all measurement systems appeared to be sufficiently reliable. Jump height values recorded varied among different technologies from trivial to moderate effect size. Sufficient concurrent validity between My Jump 2 and the TVJM ($r = 0.93$, $d = 0.15$) was found, but not between My Jump 2 and HomeCourt ($r = 0.85$, $d = 0.68$). In comparison, HomeCourt in this study was the last valid system because it registered 12.85% and 9.71% higher than the average jump heights measured by My Jump 2 and the TVJM, respectively, for the total 180 jumps.

Vertical jump tests have been investigated for more than 50 years to determine lower-body maximal explosive strength. There are various ways to define jump height. Force platform measurement [6,29] has been widely considered the gold-standard test but requires several instruments for recording and processing the data. It calculates the vertical displacement of an individual's center of mass (CoM) using the measured kinetic data. To our best knowledge, measuring flight time to determine jumping performance was first reported in the 1980s [30]. At that time, the flight time was measured by electronic apparatus, which consisted of a digital timer (±0.001 s) and a resistive platform, while the jumps were performed with hands on the hip and minimal horizontal and lateral displacements. In fact, the technology and principle behind My Jump 2, a mobile application, were similar to those of Bosco's study. My Jump 2 used video frame identification to recognize the flight time [17], the time difference between the two frames marked at take-off and landing. The mobile apparatus makes the assessment portable and user-friendly. The TVJM is a type of linear position transducer [8]. A cord is attached to a rubber mat on the ground, and the belt at the athlete's waist is pulled out of its housing when the athlete jumps. This device assumes that the change in length of the cord is the change of the

athlete's CoM away from the ground vertically, i.e., measuring how high an athlete reaches directly. AI motion tracking, a novel test technique, is adopted in the HomeCourt mobile application. It videotaped the jump and then analyzed the vertical displacement of the initial standing height and the peak jump height while no reflective marker was needed. The test requests participants to jump facing the camera in order to capture the jump in the frontal plane view. It is different from the other markerless motion capture with deep learning approaches, which quantifies the sagittal plane kinematics [31]. The developer has claimed that when used properly and under good lighting conditions, the error rate is under 5% [32]. CMJ can be assessed by numerous methods which are grounded in various techniques and technologies.

Of the three measurements, the jump heights registered were varied. My Jump 2 was identified as the reference standard for the criterion-related validity comparison [16]. Previous studies found an almost perfect agreement ($r > 0.99$) between the outputs of the jump height calculated using My Jump 2 and that of the force platform [11,12]. Although they have different sampling rates for capturing the take-off and landing frames, they adopt similar jump height calculations by measuring the flight time [11,17]. HomeCourt and the TVJM tended to have a higher mean jump height for CMJ (difference: 1.17–5.25 cm) compared with My Jump 2. This was probably due to the different technologies adopted for measurements. Particularly, the AI of HomeCourt used an object recognition technique while the TVJM measured the change of CoM directly. Noteworthy, the latter system may overestimate the jump height if the participant did not exactly land on the spot where s/he jumped. Furthermore, larger CVs in the two mobile applications were observed. The higher level of dispersion around the mean may be due to the blur of video motion caused by the current frame rate of the iPad camera (i.e., the sampling rate of the motion tracking) and the speed of the participant's motion. Due to the diverse limitations of the technologies, it is not recommended to compare the values among devices.

Although the motion-tracking technology for assessing jump height in HomeCourt has been established for years, this study was the first to evaluate its validity and reliability. HomeCourt is simple to use and reliable (good between-day reliability) for measuring jump height. However, due to its moderate within-day $ICC_{3,1}$ (0.82–0.83), the average jump height, instead of the best value, was suggested for analysis [2]. Noteworthily, the outputs need to be interpreted with caution; specifically, only values over MDC should be viewed as real changes (e.g., changes over $\pm 6.71$ cm in HomeCourt). The markerless motion-capturing approach of HomeCourt offers the potential to revolutionize scientific studies by allowing researchers and practitioners to collect data remotely outside of the laboratory. Furthermore, users of HomeCourt do not require any scientific knowledge or skills, e.g., the measurement of leg length, for CMJ assessment. Therefore, HomeCourt is practical for users with diverse backgrounds, highlighting its greater usability in sports performance measurement. The major drawback of HomeCourt would be the requirement for high-speed AI algorithms, and the iOS platform provided this application only recently. Regardless of the cons of HomeCourt, it could still be a user-friendly mobile application in the field in the present and the future.

Because of the cost of a mobile device, the TMJM is more affordable for measuring jump height. The findings evaluated it as the most reliable measurement (ICC: 0.92–0.97) and showed the best validity against a criterion measure ($r = 0.93$, $d = 0.15$) with the reference standard of My Jump 2, although its validity was questioned by McMahon et al. [8]. Moreover, it is lightweight and cost-friendly, so it can easily be implemented for on-field and remote assessment in sports and physical education; for example, athletes and students can self-monitor physical conditioning, following coaches' and teachers' advice. Such portable and mechanical devices are applicable for providing valid information for decision-making and training/teaching strategy modifications. Therefore, the usefulness of the TMJM was endorsed in this study.

Considering the reality of remote assessment in the sports field and schools, athletes and students may record their jump performance on different days within a week. Good to excellent reliability among devices was shown, referring to the high between-day ICC measures and low measurement error. It provided users the flexibility to assess at any possible time independently. Comparing previous studies, this study emphasized the measurement setup to assess the concurrent validity of the three devices. The iPads were placed on standardized foldable tablet stands with a marked angle and placement, allowing future replication. Standardized setup and equipment placement in validation studies for mobile applications are vital in order to limit the potential influence of the different positions of the smart devices [12,15]. Regarding the need for a basic understanding of the measurement devices, familiarization trials for users are suggested before fully implementing them in remote assessments. To fully implement remote assessment, inter-rater reliability should be assessed between self- and assessor-administered assessments. Given that the mentioned recommendations are followed, remote assessments are encouraged.

There are a few limitations to this study. First, iPads with a sampling frequency of 120 Hz were used in this study. These are common in schools but may not be able to identify the accurate take-off and landing frames due to the relatively low sampling frequency. More advanced devices adopting a higher sampling frequency of 240 Hz may provide even higher reliability. Second, the present study was limited to individuals with high levels of physical activity participation. Future reliability studies should include participants with different backgrounds, such as school-aged students. Third, the gold standard of jump measurement, i.e., the force platform, was not used in this study, whereas a validated vertical jump measuring device, My Jump 2, was selected as a reference standard for comparison. Future studies should incorporate this for comparison.

The study highlighted significant differences among the three measurement systems in assessing CMJ height. All three measurements were highly reliable. The remote assessment tools, HomeCourt and the TVJM, showed great agreement with the well-established assessment tool, My Jump 2. Noteworthily, the same apparatus should be used for repeated measures because of the computational differences. The consideration of average jump heights was suggested, and the high between-day reliability of all systems allows users to assess flexibly. This study also provided MDC to indicate meaningful changes. The recent technologies permit practitioners with limited training backgrounds in sports measurement to self-administer physical performance tests remotely.

**Author Contributions:** Conceptualization, G.C.-C.C.; methodology, G.C.-C.C. and W.-Y.P.; formal analysis, G.C.-C.C. and Y.-H.K.; investigation, W.-Y.P.; resources, G.C.-C.C.; data curation, W.-Y.P.; writing—original draft preparation, G.C.-C.C.; writing—review & editing, Y.-H.K.; funding acquisition, G.C.-C.C. All authors have read and agreed to the published version of the manuscript.

**Funding:** The research and the article processing charge were funded by the Research Grants Council, Hong Kong, under Research Matching Grant Scheme (CB308).

**Institutional Review Board Statement:** The study was conducted according to the guidelines of the Declaration of Helsinki and approved by the Human Research Ethics Committee of The Education University of Hong Kong (Ref. no. 2020-2021-0362, 27 May 2021l).

**Informed Consent Statement:** Informed consent was obtained from all subjects involved in the study.

**Data Availability Statement:** The data presented in this study are available on request from the corresponding author. The data are not publicly available due to privacy concern.

**Acknowledgments:** The research team would like to thank all participants for participating in the study during the pandemic.

**Conflicts of Interest:** The authors declare no conflict of interest.

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
