# Peer review of "The Concurrent Validity and Test-Retest Reliability of Possible Remote Assessments for Measuring Countermovement Jump: My Jump 2, HomeCourt & Takei Vertical Jump Meter"

_applsci, doi:10.3390/app13042142_

Round 1

Reviewer 1 Report

The Concurrent Validity and Test-retest Reliability of Possible Remote Assessments for Measuring Countermovement Jump: My Jump 2, HomeCourt & Takei Vertical Jump Meter

The purpose of the study was to investigate the concurrent validity and test-re-test reliability between three different remote assessments of CMJ. Authors showed that Takei Vertical Jump Meter was the most reliable system to calculate CMJ. The study is well designed, has a strong background, although there are a few points that needs to be addressed before the study will consider for publication.

Major comments:

1. Authors used three different remote methods to calculate CMJ height. The comparison was made to My jump 2. However, force platforms are the golden standard of measuring vertical jumps (as the authors mention inside the discussion and limitations). Then, why these remote methods did not compared to the results from a force platform? Why My Jump 2 was used as reference here (this information was presented in line 298, thus Authors should mention this earlier in the manuscript)? This really limits the generalization of the results.    

2. Authors provide only 3 CMJ during each session. Were the 3 CMJ sufficient for participants to achieve their highest vertical jump? Since this is a validity – reliability study, why Authors did not gave more trials to participants?

3. Line 31: I suggest changing “sportspeople” to “sports professionals”. Also, when authors refer to “average jump heights” they mean that coaches should measure all three CMJ and use the average height performance? If this is the case, then this should be highlighted inside the methods since it is not clear that the average performance of all three CMJ was used for the analysis.

General comments;

Abstract: Abstract is well written and provides a good overview of the study. No comments here.

Introduction:

The intro is well written, has a good flaw and presents recent and relative references.

Line 35: For all team sports, the off-season period is the hardest of all. I suggest changing the off-season to “transition phase” to avoid any clarity problems.

Materials and Methods:

Lines 84-85: How Authors used the self-reported physical activity of the participants? Why this data was collected?

Line 86: Familiarization is normally performed during different days than measurement days. This is to ensure that participants are well trained with the procedures and execution of CMJ. Are the Authors refer to familiarization as warm-up CMJ’s?

Line 90: What were the inclusion criteria for participants?

Paragraph procedures: If Authors used the average jump height form all three CMJ this is the paragraph to mention it. Also, did the participants have an instant feedback after each CMJ?

Figure 1: Very good.

Line 127: I suggest using the terms warm-up CMJs and not familiarization trials. In addition, Authors should clarify if participants visited the laboratory on a different day to be familiarized with CMJ or they already knew how to perform CMJ according to their individual IPAQ-S.

Figure 2: Regarding the TVJM, is the base attached to the floor? Or participants were forced to place their feet in such a narrow standing base? Also, I don’t think that figure 3 provides valuable information for readers. It is clear in the text.

Line 165: Please refer to the method in the first line since it needs the reader to reach the fourth line to see the HOMECOURT method.

Line 170: Explain the abbreviation AI before presenting.

Results:

Results are very good. Tables are in a good state.

Line 200: I am not sure I follow the 180-jump data meaning here. I am sure that this refers to all trials for all participants but Authors should clearly state this for readers.

Discussion:

Discussion is very good. A suggestion here is to present a take home message at the end of each paragraph in order to summarize the meanings for readers.

Lines 285-289: Should this be considered as a limitation for the application of My Jump 2?

Line 323: Authors have not presented IPAQ-S results. Consequently, readers don’t know the level of physical fitness of the participants.  

Well done.

Reviewer 2 Report

This study examined the concurrent validity 15 and test-retest reliability of three portable measurement systems for CMJ. It would be better if some changes based on the comments are corrected and revised. 

1. Introduction

- Last paragraph: It should be a core paragraph focus on describing purpose statements. However, it mixed up with both the value of study and the the purpose of statements including the hypothesis of study. Please divide into each paragraph.

2.  Methods

- Statistical Anal.: Although normality test using Shapiro-Wilk test and Repeated measure ANOVA were indicated, the outcomes with output were omitted in Results. It just described simply in 1st sentence in Results part.

3. Results

- The ANOVA with repeated measures output should be presented with Table. Readers would wonder how the outcomes were differ by over-time and among the three systems.
